# Estimation of Lifetime QALYs Based on Lifestyle Behaviors

**DOI:** 10.3390/ijerph18199970

**Published:** 2021-09-22

**Authors:** Shinichi Noto, Shota Saito, Takeru Shiroiwa, Takashi Fukuda

**Affiliations:** 1Department of Rehabilitation, Niigata University of Health and Welfare, Niigata 950-3198, Japan; 2Department of Medical Informatics and Statistics, Niigata University Graduate School of Medical and Dental Sciences, Niigata 950-2181, Japan; ssaito@med.niigata-u.ac.jp; 3Center for Outcomes Research and Economic Evaluation for Health (C2H), National Institute of Public Health, Wako 351-0197, Japan; t.shiroiwa@icer.jp (T.S.); t-fukuda@niph.go.jp (T.F.)

**Keywords:** lifestyle behavior, health-related quality of life, QALY, physical activity, sleep

## Abstract

Various lifestyle behaviors have been known to affect health-related quality of life (HRQL) and life expectancy. However, the impact on quality-adjusted life years (QALYs), which can be used for health economics, has not been clarified. The purpose of this study was to estimate the impact of lifestyle behaviors on lifetime QALYs. We first examined the relationship between lifestyle behaviors and HRQL as measured by the EQ-5D-5L among 4000 participants via a web-based survey. The results of multiple regression analysis showed that physical activity and sleep were significantly related to HRQL. Therefore, we used microsimulation to estimate QALYs from physical activity and sleep, which were determined to be significant in the regression analysis. The results showed that there was a difference of 3.6 QALYs between the recommended lifestyle scenario (23.4 QALYs; 95%CI 3.6 to 35.1) and the non-recommended lifestyle scenario (19.8 QALYs; 95%CI 3.1 to 31.6). This difference was greater in the younger age group than in the older age group. The results also indicated a large difference in QALYs between physical activity and sleep. These findings may provide a significant suggestion for future health promotion measures.

## 1. Introduction

Various lifestyle behaviors are known to affect people’s health, health-related quality of life (HRQL), and life expectancy. It is our responsibility as researchers involved in public health to suggest lifestyle behaviors to maintain health and HRQL. In a large Swedish study, Ali et al. [1] showed that various lifestyle behaviors affect quality of life. They found that 30 min of daily physical activity, normal weight BMI, fruit consumption, smoking cessation, and alcohol abstinence were associated with higher HRQL, as measured by the EQ-5D. Choi et al. [2] followed the effects of moderate or vigorous physical activity (MVPA) on quality of life among elderly women in the United Kingdom for seven years and found that regular MVPA prevented a decline in quality of life. In our study [3], which surveyed a community sample with health checkups, we demonstrated that moderate physical activity and good sleep also lead to a higher quality of life. Li et al. [4] identified the following lifestyle factors affecting life expectancy in middle-aged and older adults: never smoking, body mass index of 18.5–24.9, ≥30 min/day of moderate to vigorous physical activity, moderate alcohol intake, and high diet quality score (>40% for Alternative Healthy Eating Index). Efforts are being made to use these factors in treatment to extend life expectancy [5].

On the other hand, it is also well known that deteriorating health conditions lead to disease, which in turn leads to increased medical and nursing care costs. As the impact of these various lifestyle habits on health and quality of life became clearer, several intervention studies were conducted on prominent individuals with obesity [6,7], cardiovascular disease [8], and type 2 diabetes [9]. Furthermore, although various lifestyles are known to affect people’s health, quality of life, and life expectancy, few papers have examined their economic impact [10,11]. To examine the economic impact, it is necessary to measure HRQL with a preference-based index and calculate quality-adjusted life years (QALYs). A systematic review of the effects of lifestyle interventions reported that the QALY gain from such interventions was very small at 0.003 [12]. The economic impact of health interventions has been an important concern for countries with aging societies, and there are high expectations for research using microsimulation, which simulates the impact on society from individual-level data [13]. In Japan, microsimulation has been used to calculate QALYs for vaccines [14] and cancer screening [15], and its usefulness has been confirmed. The purpose of this paper is to estimate QALYs using microsimulation from the influence of lifestyle behaviors that can be relatively improved, as has been determined by our previous research.

## 2. Materials and Methods

### 2.1. Study Design and Participants

We conducted a survey of the general Japanese population, in which they were asked to complete a web-based questionnaire. The survey was administered by Rakuten Insight, Inc. in January 2021. The target sample size of this study was 4000 participants. This research company has approximately 2.2 million panelists throughout Japan, and 4000 were recruited from that target population on a first-come, first-served basis. The participants were aged 19–89 years and were assigned to each of the eight regions (Hokkaido, Tohoku, Kanto, Chubu, Kinki, Chugoku, Shikoku, and Kyushu) according to their population. We ensured that the sample was representative of the Japanese population in terms of age, sex, and residential area during the collection phase. Participants responded to questions about household income, employment status, education, marital status, subjective symptoms, and self-reported chronic diseases.

### 2.2. Questionnaires

Assessed through self-report, participants responded to questions about the following lifestyle behaviors using Likert scales. On a three-point Likert scale, participants responded to their frequency of alcohol intake (1 = Never (no alcohol intake at all), 2 = Sometimes (2–3 times a week), 3 = Habitually) and smoking (1 = Never (have never smoked at all), 2 = Active (smoke on a daily basis), 3 = Former smoker). On a two-point Likert scale, participants responded to lack of exercise (1 = No, 2 = Yes) and sleep (1 = Well, 2 = Lack of sleep). On a four-point Likert scale, participants responded to their frequency of physical activity (1 = Very often, 2 = Usually, 3 = Not so active, 4 = Too little) and exercise habit (1 = Almost daily, 2 = 3–5 times/week, 3 = 1–2 times/week, 4 = Too little). Physical activity comprises subjective activity in daily life, while exercise habit comprises sweaty exercise for more than 20 min.

The questionnaire included a lifestyle survey and an EQ-5D-5L. The EQ-5D-5L is a generic preference-based measure of HRQL developed by the EuroQol Group [16]. It includes five dimensions: mobility (MO), self-care (SC), usual activities (UA), pain/discomfort (PD), and anxiety/depression (AD). Each dimension has five levels. The Japanese version of the EQ-5D-5L was used in this study; therefore, the responses obtained were converted to HRQL scores based on Japanese value sets [17].

### 2.3. Statistical Analysis

A summary of HRQL scores was calculated based on gender, age group (19–29, 30–39, 40–49, 50–59, 60–69, and ≥70–year–old), and lifestyle factors. These scores were compared using variance analysis. To detect the influence of demographic and lifestyle factors on the HRQL scores, these variables were added to an analysis of variance (ANOVA). The influence of demographic characteristics and lifestyle factors on HRQL was determined using multiple regression analysis. Sex, age, drinking, smoking, physical activity, and sleep, with the relevant dummy variables, were included as covariates in a multiple regression model with the total HRQL score as the outcome.

We performed a microsimulation to estimate the lifetime expected QALYs for our sample [18]. The microsimulation model addresses a limitation of the deterministic cohort model because it can more easily estimate expected future outcomes based on individuals’ characteristics at baseline. In a microsimulation model, outcomes are generated for each individual and are used to estimate the distribution of an outcome for a sample of potentially heterogeneous individuals.

In this simulation, we constructed a two-state model, which included living and dead states to estimate lifetime QALYs, and created a hypothetical cohort including 4000 subjects based on the demographic statistics of our sample. Then, using age, gender, and only the significant factors from the regression model results, we calculated the expected HRQL score for each participant. We assumed that lifestyle factors were generated independently and that, for each subject, those factors never changed until death. Long-term survival was modelled using the Japanese life table in 2018. Mortality was dependent on the age and gender of each subject. In the microsimulation, QALYs were discounted at a rate of 2% per year.

The significance level was set at 0.05. Statistical analyses and microsimulation were performed using STATA 15.0 and TreeAge Pro 2021 R1.1, respectively. This study was approved by the Ethics Committee of Niigata University of Health and Welfare (No. 18567–210114). Informed consent was obtained from all individual participants included in the study.

## 3. Results

### 3.1. Characteristics of the Participants

Table 1 shows the demographic and health characteristics of the participants. Of the total, 50.4% were female subjects (mean age = 49.8 years). Concerning the participants’ subjective symptoms, 36.1% reported shoulder stiffness, 28.7% reported lower back pain, and 32.3% did not report any symptoms. Among self-reported diseases, 726 participants (18.2%) reported having hypertension, 268 (6.7%) reported orthopedic disorder, 222 (5.6%) reported having diabetes, and 2461 participants (61.5%) did not report any disease. Self-reported lifestyle habits are shown in Table 2 along with the results of the EQ-5D-5L.

### 3.2. EQ-5D-5L Scores

Table 2 shows the results of the EQ-5D-5L scores for each factor. No differences were found based on gender and age. However, in the comparison of lifestyle behaviors, significant differences were found in all factors: drinking (*p* = 0.039), smoking (*p* = 0.022), lack of exercise (<0.001), physical activity (<0.001), exercise habits (<0.001), and sleeping (<0.001). In the case of physical activity, the score of participants who answered “very often” was 0.936 ± 0.085, while that of participants who answered “too little” was 0.879 ± 0.188. Concerning sleep, participants who reported to have slept “well” scored 0.921 ± 0.114, while the score for those who reported experiencing “lack of sleep” was lower, at 0.853 ± 0.148.

### 3.3. Regression Analysis

Table 3 shows the results of the multiple regression analysis. Differences were found at all levels of physical activity, with –0.026 (*p* = 0.002) for “usually”, –0.053 (<0.001) for “not so active”, and –0.084 (<0.001) for “too little” compared with “very often”. Concerning sleep, compared with “lack of sleep,” the coefficient for “good” was 0.063. There were no significant differences in the factors of drinking, smoking, and lack of exercise.

### 3.4. Microsimulation for Estimating Lifetime QALYs

QALYs were estimated by applying two factors: physical activity and sleep. These were statistically significant in the multiple regression analysis and significant in our previous study [15] on a cohort of 4000 people. As shown in Table 4, the highest number of 23.4 (95%CI 3.6 to 35.1) QALYs was obtained when physical activity was “very often” and participants slept “well”. However, those with “too little” physical activity and “lack of sleep” gained the fewest QALYs (19.8 QALYs; 95%CI 3.1 to 31.6). The difference from the base case was 1.7 QALYs and –1.9 QALYs, respectively, resulting in a difference of 3.6 QALYs. In addition, the age-specific analysis showed that the difference between the recommended and non-recommended lifestyle behaviors was 5.5 QALYs among the 10 year olds and 1.8 QALYs among the 70 year olds (Figure 1). The 95% CIs for lifetime expected QALYs are presented in Appendix A.

## 4. Discussion

We obtained lifestyle and EQ-5D-5L scores based on a large web-based survey. The updated values were comparable and consistent when compared with our previous survey of health check-ups in 2015 [3]. Compared with the standard values for each age group [19], our survey method seems to be reproducible and reliable. Therefore, we believe that the calculation of QALYs using physical activity and sleep as variables, which was also significant in this study, is robust and can contribute to future health economic evaluation and policy making.

Comparing various lifestyle behaviors and HRQL scores, the impact of alcohol consumption and smoking on HRQL was lesser compared with the impact of physical activity, exercise habits, and sleep status. These results are consistent with our previous study [3] and recent surveys [20,21], and are, therefore, valid. A comparison of the EQ-5D-5L scores for alcohol consumption and smoking revealed that the difference between those who said they never consumed alcohol and those who said they habitually consumed alcohol was 0.013, whereas the difference between smokers and nonsmokers was 0.016. Ali et al. [1] examined the effect of alcohol consumption and smoking on quality of life as measured by the EQ-5D-3L in Sweden and reported only a difference of 0.03 for each, making their effect on quality of life smaller than other factors. A large study [22] conducted in England also found that alcohol consumption did not have a significant impact on quality of life as measured by the EQ-5D. Moreover, a study in Finland [23] reported that the relationship between alcohol consumption and quality of life was not clear, and that the health benefits of moderate drinking were ambiguous. Conversely, the relationship between smoking and quality of life has been unclear in recent papers [24,25]. In our study, we did not find a significant difference in smoking among the former smokers compared to the active smokers. Therefore, it was assumed that future smoking cessation would not significantly affect QALYs in the framework of this study and was not included in the result. Therefore, the results of our survey seem to be generally consistent.

Among the lifestyle behaviors that made a significant difference in EQ-5D-5L scores was physical activity, which was 0.09 higher for those who were active very often compared with those who were relatively not active. For sleep, participants who reported good sleep presented 0.068 higher scores than those who reported lack of sleep. These factors, therefore, seem to have a greater impact on quality of life than factors of alcohol consumption and smoking.

Based on the above results, we attempted to estimate QALYs by taking lifestyle behaviors into account. We found that there was a difference of 3.6 QALYs between those who maintained a state of good sleep and were active very often, and those who experienced lack of sleep and performed very little activity. It should be noted that this is not a mere extension of life expectancy but QALYs that take HRQL into account. In the evaluation of medical technology, QALYs are treated as an outcome in many countries, including the United Kingdom [26], Australia [27], and Canada [28] (as is also the case in Japan [29]). They are regarded as an important indicator. Therefore, it is very meaningful that we were able to derive this value in our study. In the future, the evaluations of the cost-effectiveness of health programs should be conducted, and our research results would serve as an anchor for such evaluations.

This is not to say that there are no studies that use the impact of such lifestyle factors to estimate QALYs. Barbosa et al. [30] used a Markov model to calculate QALYs based on differences in drinking patterns and conducted a cost-effectiveness analysis. Xu et al. [31] calculated QALYs for the effects of smoking. They found that male cigarette smokers aged 25–29 years lost 8.1 QALYs compared to those who never used tobacco. Males who are current smokeless tobacco users, aged 25–34 years, lost 4.1 QALYs. Although there are studies that examine the impact of a single factor on QALYs, our study is significant because it examines multiple factors.

However, there are certain limitations to this study. Not only lifestyle, but subjective symptoms and prevalent diseases also influence QALYs. In fact, they should have a greater impact on QALYs. This study could not consider demographic characteristics such as income, education, and occupation, as well as environmental influences such as urbanization and exposure to pollution. In addition, only a limited number of lifestyle factors predicted the calculation of QALYs. In fact, physical activity, smoking, sleep, diet, and other factors may have complex confounding effects on health status, so the results of this study may be of limited interpretation. Since these factors may also have certain effects on health, we believe that they should be reflected in future studies. Another limitation of this web survey is that we recruited a fixed upper limit of participants. Therefore, we cannot deny the possibility of selection bias in that highly motivated respondents were gathered. Since selection bias has a significant impact on the results of a survey, the results of this study may also not be based on a representative sample of the population. Future research that takes these influences into account is required.

## 5. Conclusions

We investigated the relationship between lifestyle behavior and health-related quality of life using a web-based survey and identified physical activity and sleep as significantly related factors that affect quality of life. We also estimated QALYs by applying microsimulation to these two factors. The results showed that there was a difference of 3.6 QALYs between the recommended and non-recommended lifestyle scenarios. Estimating QALYs by considering the effect of lifestyle behaviors is expected to be useful for future health promotion measures.

## Figures and Tables

**Figure 1 ijerph-18-09970-f001:**
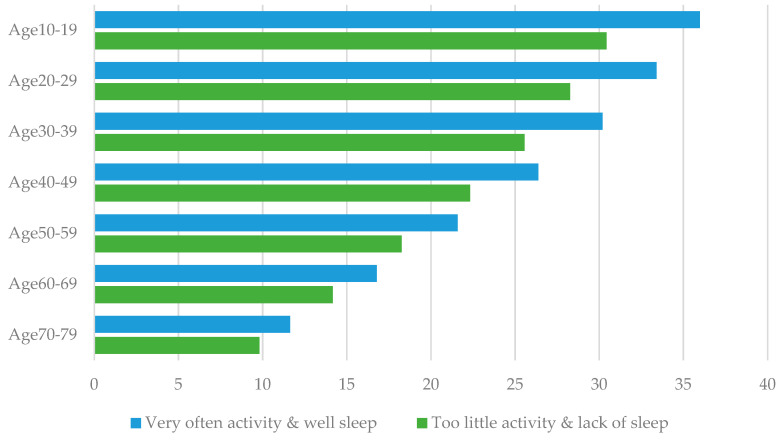
Comparison of the difference in QALYs between the recommended and non-recommended lifestyle behaviors.

**Table 1 ijerph-18-09970-t001:** Demographic factors of the study participants.

		*n*	%
Gender	Male	1985	49.6
	Female	2015	50.4
Age	19–29	611	15.3
	30–39	597	14.9
	40–49	771	19.3
	50–59	681	17.0
	60–69	676	16.9
	>70	664	16.6
Region of Residence	Hokkaido	168	4.2
	Tohoku	340	8.5
	Kanto	1415	35.4
	Chubu	647	16.2
	Kinki	654	16.4
	Chugoku	225	5.6
	Shikoku	114	2.9
	Kyushu	437	10.9
Employment	Full-time worker	1648	41.2
	Part-timer	742	18.6
	Self-employed	247	6.2
	Homemaker	609	15.2
	Retired	248	6.2
	Leave	418	10.4
	Others	88	2.2
Marital status	Married	2527	63.2
	Unmarried	1071	26.8
	Divorced or bereaved	402	1.0
Education	Junior highschool	83	2.1
	highschool	1187	29.7
	College etc.	878	22.0
	University	1841	46.0
	Graduate school	11	0.3
Household income	<JPY 2mil	342	8.6
	JPY 2 mil ≤ 4 mil	864	21.6
	JPY 4 mil ≤ 6 mil	808	20.2
	JPY 6 mil ≤ 10 mil	864	21.6
	JPY 10 mil ≤ 15 mil	304	7.6
	≥JPY 15 mil	84	0.2
	Refused, unknown	734	18.4

**Table 2 ijerph-18-09970-t002:** EQ-5D-5L score based on characteristics and lifestyle behaviors of the study participants.

		*n*	EQ-5D-5L Index Score (Mean ± SD)	*p* Value	EQ-5D-5L VAS (Mean ± SD)	*p* Value
	All	4000	0.891 ± 0.134		79.4 ± 17.0	
Gender	Male	1985	0.890 ± 0.145	0.598	79.5 ± 16.7	0.573
	Female	2015	0.892 ± 0.123		79.2 ± 17.3	
Age	19–29	611	0.882 ± 0.151	0.335	77.7 ± 18.0	<0.001
	30–39	597	0.894 ± 0.116		78.0 ± 17.0	
	40–49	771	0.890 ± 0.136		76.9 ± 18.8	
	50–59	681	0.891 ± 0.132		79.3 ± 17.4	
	60–69	676	0.899 ± 0.122		81.1 ± 15.0	
	>70	664	0.890 ± 0.145		83.3 ± 14.4	
Drinking	Never	1522	0.884 ± 0.142	0.039	78.4 ± 18.0	0.014
	Sometimes	1399	0.894 ± 0.128		80.4 ± 15.2	
	Habitually	1041	0.897 ± 0.131		79.5 ± 17.2	
Smoking	Never	2290	0.896 ± 0.128	0.022	79.9 ± 16.7	0.007
	Active	666	0.880 ± 0.141		77.5 ± 18.8	
	Former smokers	1006	0.887 ± 0.144		79.3 ± 16.5	
Lack of Exercise	No	839	0.915 ± 0.141	<0.001	85.2 ± 14.3	<0.001
	Yes	3161	0.885 ± 0.132		77.8 ± 17.3	
Physical Activity	Very often	390	0.936 ± 0.085	<0.001	86.8 ± 12.6	<0.001
	Usually	1653	0.909 ± 0.113		82.1 ± 14.4	
	Not so active	1285	0.879 ± 0.130		77.1 ± 16.7	
	Too little	672	0.846 ± 0.188		72.7 ± 21.9	
Exercise habit	Almost daily	366	0.921 ± 0.105	<0.001	84.5 ± 14.8	<0.001
	3–5/week	566	0.911 ± 0.105		83.8 ± 13.9	
	1–2/week	846	0.891 ± 0.121		80.6 ± 14.4	
	Too little	958	0.894 ± 0.125		78.7 ± 16.6	
	Almost never	1264	0.871 ± 0.162		75.7 ± 19.7	
Sleeping	Well	2234	0.921 ± 0.114	<0.001	83.9 ± 13.7	<0.001
	Lack of sleep	1766	0.853 ± 0.148		73.7 ± 19.0	

**Table 3 ijerph-18-09970-t003:** Relationship between EQ-5D-5L scores, demographic characteristics, and lifestyle behaviors.

		EQ-5D-5L		
		Coefficient	95% CI	*p* Value
	Intercept	0.871	0.847–0.895	<0.001
Gender	Male	–0.004	–0.013–0.004	0.315
	Female	–	–	–
Age	19–29	–	–	–
	30–39	0.020	0.005–0.035	0.007
	40–49	0.018	0.006–0.032	0.014
	50–59	0.017	0.002–0.031	0.025
	60–69	0.015	0.001–0.030	0.040
	>70	–0.003	–0.018–0.012	0.698
Drinking	Never	–	–	–
	Sometimes	0.006	–0.004–0.016	0.223
	Habitually	0.007	–0.003–0.019	0.156
Smoking	Never	0.013	0.001–0.025	0.033
	Active	–	–	–
	Former smokers	0.004	–0.008–0.017	0.505
Lack of Exercise	No	0.006	–0.006–0.018	0.349
	Yes	–	–	–
Physical Activity	Very often	–	–	–
	Usually	–0.026	–0.042–−0.010	0.002
	Not so active	–0.053	–0.072–−0.035	<0.001
	Too little	–0.084	–0.106–−0.064	<0.001
Exercise habit	Almost daily	–	–	–
	3–5/week	0.000	–0.017–0.017	0.993
	1–2/week	–0.009	–0.027–0.008	0.304
	Too little	0.005	–0.013–0.023	0.590
	Almost never	0.001	–0.018–0.020	0.909
Sleeping	Well	**0.063**	**0.055–0.072**	**<0.001**
	Lack of sleep	–	–	–

Adjusted R2: 0.098 for EQ-5D-5L. “–” shows reference group. CI: confidence interval.

**Table 4 ijerph-18-09970-t004:** Lifetime expected QALYs per person in scenario analysis.

QALYs per Person			Physical Activity			
			Very Often	Usually	Not So Active	Too Little
All						
Sleeping	Well	Expected value	23.4	22.7	22.1	21.3
		Difference from base case	1.7	1.0	0.4	–0.4
	Lack of sleep	Expected value	21.8	21.2	20.5	19.8
		Difference from base case	0.1	–0.5	–1.2	–1.9
Age 10–19						
Sleeping	Well	Expected value	36.0	35.0	34.0	32.8
		Difference from base case	2.6	1.7	0.6	–0.5
	Lack of sleep	Expected value	33.6	32.6	31.6	30.4
		Difference from base case	0.8	–0.7	–1.7	–2.9
Age 20–29						
Sleeping	Well	Expected value	33.4	32.5	31.6	30.5
		Difference from base case	2.4	1.5	0.6	–0.5
	Lack of sleep	Expected value	31.2	30.3	29.4	28.3
		Difference from base case	0.2	–0.7	–1.6	–2.7
Age 30–39						
Sleeping	Well	Expected value	30.2	29.4	28.5	27.6
		Difference from base case	2.2	1.4	0.5	–0.5
	Lack of sleep	Expected value	28.2	27.4	26.5	25.6
		Difference from base case	0.2	–0.6	–1.5	–2.4
Age 40–49						
Sleeping	Well	Expected value	26.4	25.7	24.9	24.1
		Difference from base case	1.9	1.2	0.5	–0.4
	Lack of sleep	Expected value	24.6	23.9	23.2	22.3
		Difference from base case	0.2	–0.5	–1.3	–2.1
Age 50–59						
Sleeping	Well	Expected value	21.6	21.0	20.4	19.7
		Difference from base case	1.6	1.00	0.4	–0.3
	Lack of sleep	Expected value	20.2	19.6	19.0	18.3
		Difference from base case	0.1	–0.5	–1.1	–1.8
Age 60–69						
Sleeping	Well	Expected value	16.8	16.3	15.8	15.3
		Difference from base case	1.2	0.8	0.3	–0.3
	Lack of sleep	Expected value	15.7	15.2	14.7	14.2
		Difference from base case	0.1	–0.3	–0.8	–1.4
Age 70–79						
Sleeping	Well	Expected value	11.6	11.3	11.0	10.6
		Difference from base case	0.9	0.6	0.2	–0.2
	Lack of sleep	Expected value	10.9	10.5	10.2	9.8
		Difference from base case	0.1	–0.2	–0.6	–1.0

## Data Availability

Data is contained within the article or Appendix A.

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
