# Peer review of "Estimation of Lifetime QALYs Based on Lifestyle Behaviors"

_ijerph, 2021, doi:10.3390/ijerph18199970_

Round 1

Reviewer 1 Report

This is essentially a methodological study: it uses a web-based survey to estimate relationships between self-reported behaviors and health-related quality of life and then uses these findings to conduct a micro-simulation to project quality adjusted life years for different amounts of sleep quality and exercise patterns. I would think these findings would be helpful for Japanese medic-economics researchers who wish to place a value on changes in lifestyle. There are several pieces that would make this paper stronger:

1) in the introduction, there should be more evidence for demand for QALY projections for the joint impacts of exercise patterns and sleep quality.....I am having a hard time thinking of situations where another researcher would use the findings in table 4. It would also be helpful to speak in more detail about the micro-simulation strategy adopted and how this approach has and has not been valuable in related contexts.

2) The authors did not include questions or data on other demographic features (income, education, occupational type, other sources of stress and well-being) and environmental factors (urbanization, pollution exposure).....the article should justify these gaps in some detail since these are factors with strong influence on HRQoL and longevity. The methods section should say a bit more about the details of the microsimulation" the referenced textbook describes multiple strategies and critically appraises them

3) The authors do not say anything about response rates (and how they differed by age and health status groups) and what if any biases may have resulted from the web-based survey approach.

5) The authors do not make a convincing case for not including smoking in their projections....I believe there is strong evidence for smoking impacts on longevity and the discussion highlights studies on smoking HRQoL and longevity.

6) The regression analyses examine each lifestyle factor's impacts on HRQoL controlling for demographic but not for other lifestyle behaviors. Since PA, smoking, sleep and diet have been found to be correlated in complex ways in multiple contexts, it is unclear that projections building on the bivariate relationships between lifestyle factors and HRQoL adequately model these relationships. The authors could not that this is unexplored and would be helpful. There is a paragraph in the discussion that seems to be speaking to these limitations but it is not clear.

5) Please reconsider the use of "good" and "bad" lifestyle throughout the text. These seem like value judgements and are not descriptive.....it is too easy to go from a paper with this framing to blaming a person with a high stress high physical labor job who does not exercise for their outcomes. Even calling the preferred lifestyle behaviors "healthier" is still a value judgement.....how about calling these "recommended lifestyle" or similar more neutral term. 

Author Response

Dear Reviewer 1

We would like to thank the reviewer for taking time out of his busy schedule to give us suggestions that will enhance the value of our paper. In response to your suggestion, we have carefully revised the paper again and would highly appreciate it if you could review it again.

Point 0: This is essentially a methodological study: it uses a web-based survey to estimate relationships between self-reported behaviors and health-related quality of life and then uses these findings to conduct a micro-simulation to project quality adjusted life years for different amounts of sleep quality and exercise patterns. I would think these findings would be helpful for Japanese medic-economics researchers who wish to place a value on changes in lifestyle. There are several pieces that would make this paper stronger:

Response 0: Thank you very much for your kind support during such a busy time. I would like to express my sincere gratitude. We have taken your suggestion seriously and have made all possible revisions.

Point 1: in the introduction, there should be more evidence for demand for QALY projections for the joint impacts of exercise patterns and sleep quality.....I am having a hard time thinking of situations where another researcher would use the findings in table 4. It would also be helpful to speak in more detail about the micro-simulation strategy adopted and how this approach has and has not been valuable in related contexts.

Response 1: We have added the following sentences to the introduction according to the points you raised.

A systematic review of the effects of lifestyle interventions reported that the QALY gain from such interventions was very small at 0.003 [12]. The question of how much economic impact health interventions can have is an important concern for countries with aging societies, and there are high expectations for research using microsimulation, which simulates the impact on society from individual-level data [13]. In Japan, microsimulation has been used to calculate QALYs for vaccines [14] and cancer screening [15], and its usefulness has been confirmed. The purpose of this paper is to estimate QALYs using microsimulation from the influence of lifestyle behaviors that can be relatively improved, based on our previous research.

  1. Vargas-Martínez AM, Romero-Saldaña M, De Diego-Cordero R. Economic evaluation of workplace health promotion interventions focused on Lifestyle: Systematic review and meta-analysis. J Adv Nurs. 2021 Sep;77(9):3657-3691. doi: 10.1111/jan.14857.
  2. C. Government of Canada, Microsimulation approaches. Strengths and drawbacks., (2009). https://www.statcan.gc.ca/eng/microsimulation/modgen/new/chap1/chap1-4 (accessed Aug 26, 2021).
  3. Igarashi A, Hirose E, Kobayashi Y, Yonemoto N, Lee B. Cost-effectiveness analysis for PCV13 in adults 60 years and over with underlying medical conditions which put them at an elevated risk of pneumococcal disease in Japan. Expert Rev Vaccines. 2021 Jul 30:1-13. doi: 10.1080/14760584.2021.1952869.
  4. Huang HL, Leung CY, Saito E, Katanoda K, Hur C, Kong CY, Nomura S, Shibuya K. Effect and cost-effectiveness of national gastric cancer screening in Japan: a microsimulation modeling study. BMC Med. 2020 Sep 14;18(1):257. doi: 10.1186/s12916-020-01729-0.

Point 2: The authors did not include questions or data on other demographic features (income, education, occupational type, other sources of stress and well-being) and environmental factors (urbanization, pollution exposure).....the article should justify these gaps in some detail since these are factors with strong influence on HRQoL and longevity. The methods section should say a bit more about the details of the microsimulation" the referenced textbook describes multiple strategies and critically appraises them.

Response 2: Thank you very much for pointing it out. As you pointed out, we were not able to fully examine the demographic characteristics in this study. This has been added to the discussion. In addition, microsimulation has been described in more detail as follows.

This study could not consider demographic characteristics such as income, education, and occupation, as well as environmental influences such as urbanization and exposure to pollution. Since these factors may also have certain effects on health, we believe that they should be reflected in future studies.

The microsimulation model addresses a limitation of the deterministic cohort model because they can more easily estimate expected outcomes in the future based on the individuals’ characteristics at baseline. In a microsimulation model, outcomes are generated for each individual and are used to estimate the distribution of an outcome for a sample of potentially heterogeneous individuals.

In this simulation, we constructed a two-state model, which including alive and dead states, and created a hypothetical cohort including 4000 subjects based on the demographic statistics of our sample.

Point 3: The authors do not say anything about response rates (and how they differed by age and health status groups) and what if any biases may have resulted from the web-based survey approach.

Response 3: Thank you very much for pointing it out. In this study, a web survey was conducted by recruiting the upper limit as 4000. Therefore, the response rate was not tabulated. However, as you pointed out, there is a possibility that selection bias may have affected the results, and we have added this in the discussion.

In addition, the web survey recruited participants with a fixed upper limit. Therefore, we cannot deny the possibility of a selection bias that motivated respondents were attracted.

Point 4: The authors do not make a convincing case for not including smoking in their projections....I believe there is strong evidence for smoking impacts on longevity and the discussion highlights studies on smoking HRQoL and longevity.

Response 4: We take your comments seriously and have added the following in the discussion.

In our study, we did not find a significant difference in smoking among the former smokers compared to the active smokers in this study. Therefore, it was assumed that future smoking cessation would not significantly affect QALYs in the framework of this study and was not included in the results.

Point 5: The regression analyses examine each lifestyle factor's impacts on HRQoL controlling for demographic but not for other lifestyle behaviors. Since PA, smoking, sleep and diet have been found to be correlated in complex ways in multiple contexts, it is unclear that projections building on the bivariate relationships between lifestyle factors and HRQoL adequately model these relationships. The authors could not that this is unexplored and would be helpful. There is a paragraph in the discussion that seems to be speaking to these limitations but it is not clear.

Response 5: Thank you very much for your comments. Regarding the limitations of this study, we have added a comment in the discussion related to point 4 above.

In addition, only a limited number of lifestyle factors predicted the calculation of QALYs. In fact, physical activity, smoking, sleep, diet, and other factors may have complex confounding effects on health status, so the results of this study may be of limited interpretation.

Point 6: Please reconsider the use of "good" and "bad" lifestyle throughout the text. These seem like value judgements and are not descriptive.....it is too easy to go from a paper with this framing to blaming a person with a high stress high physical labor job who does not exercise for their outcomes. Even calling the preferred lifestyle behaviors "healthier" is still a value judgement.....how about calling these "recommended lifestyle" or similar more neutral term.

Response 6: We sincerely accepted and understood your point. We are very grateful for your suggestion, as it matches the image we had in mind. Therefore, we have changed the expressions "good" and "bad" in the abstract and text to "recommended" and "not recommended" lifestyle.

Reviewer 2 Report

General comments:

The paper is in general well written and appears to be a follow-up on a previous study. The sample size is large. However, I think that in particular the abstract and introduction need to be adjusted to better describe the results- in the end, the authors only calculated QALYs for sleep and physical activity, but make it sound like they mean to calculate QALYs for all lifestyle factors. I do also have some concerns about the methodology- there are a lot of assumptions made that current lifestyle choices are unchanging. In general, I would expect that poor lifestyle choices are likely correlated and also may have interactive effects (e.g. unhealthy diet and smoking). Finally, the authors could do a better job indicating what this study adds to the literature. Their discussion seems to imply this study only verifies previous studies showing the same thing.

Abstract:

More information is needed in the abstract about the methods. Where and how were these participants recruited? We should at least know what country they are from and how the participants were identified. Was there an age restriction? Also you mention “younger age group” but this it is never defined what this means.

Similarly, how do you define “best” and “worst” lifestyle? It is hard to interpret these results.

I would also suggest that you adjust the precision of the QALY estimates- I am doubtful that it is appropriate to report the results to three decimal points. Related, are there 95% confidence interval results associated with these estimates?

Introduction:

Line 27: who is “our” in this line? Whose responsibility exactly is it to propose lifestyle factors? The public health community?

Line 51: as the guiding research question for the entire paper, this is nonspecific. Which lifestyle factors? How were they selected?

Methods:

Line 57: was a sample size calculation performed? If so, what were the baseline assumptions?

Line 55: you say the survey was web-based; was it administered through email? If so, how were emails obtained? It is important to understand recruitment methods to understand whether there is selection bias.

Lines 66-81: were you asking these questions only about what they currently do, or also what they did in the past?

Results:

Line 141: it is not until here that I understood you estimated QALYs only for sleep and physical activity. This should be made more clear in the abstract, and also the overall aims of the paper (you did not aim to estimate QALYs for all lifestyle factors, but only for ones found to be significant).

Discussion:

Lines 158-161: it is good that your results were consistent with previous studies, but then what value is added by this new study?

Author Response

Dear Reviewer 2

We would like to thank the reviewer for taking time out of his busy schedule to give us suggestions that will enhance the value of our paper. In response to your suggestion, we have carefully revised the paper again and would highly appreciate it if you could review it again.

Point 0: The paper is in general well written and appears to be a follow-up on a previous study. The sample size is large. However, I think that in particular the abstract and introduction need to be adjusted to better describe the results- in the end, the authors only calculated QALYs for sleep and physical activity, but make it sound like they mean to calculate QALYs for all lifestyle factors. I do also have some concerns about the methodology- there are a lot of assumptions made that current lifestyle choices are unchanging. In general, I would expect that poor lifestyle choices are likely correlated and also may have interactive effects (e.g. unhealthy diet and smoking). Finally, the authors could do a better job indicating what this study adds to the literature. Their discussion seems to imply this study only verifies previous studies showing the same thing.

Response 0: Thank you very much for taking time out of your busy schedule to review our paper. We have taken all the suggestions sincerely as they will improve the value of our paper. Therefore, we have revised the paper as carefully as possible and would appreciate your confirmation.

Abstract:

Point 1:More information is needed in the abstract about the methods. Where and how were these participants recruited? We should at least know what country they are from and how the participants were identified. Was there an age restriction? Also you mention “younger age group” but this it is never defined what this means.

Response 1: Thank you very much for your remarks about the abstract. We take your suggestion seriously and have revised the text as follows.

First, we examined the relationship between lifestyle behaviors and HRQL, as measured by EQ-5D-5L, through a web-based questionnaire among 4000 Japanese participants recruited through a research company.

This difference was greater in the younger age group than in the older age group.

Point 2: Similarly, how do you define “best” and “worst” lifestyle? It is hard to interpret these results.

Response 2: Thank you very much for your suggestion. Reviewer 1 also pointed out the same thing, so we have changed the expressions "good" and "bad" to "recommended" and "not recommended" lifestyle.

Point 3: I would also suggest that you adjust the precision of the QALY estimates- I am doubtful that it is appropriate to report the results to three decimal points. Related, are there 95% confidence interval results associated with these estimates?

Response 3: Thank you very much for your suggestion. We have corrected the QALYs estimates in Table 5 to report all results to two decimal points. We have also added the 95% CIs as supplement material because it would be very complicated to include them in Table 5.

Introduction:

Point 4: Line 27: who is “our” in this line? Whose responsibility exactly is it to propose lifestyle factors? The public health community?

Response 4: Thank you very much for your suggestion. Our use of the word "our" was ambiguous, so we have revised it as follows.

It is our responsibility as researchers involved in public health to suggest lifestyle behaviors to maintain health and HRQL.

Point 5: Line 51: as the guiding research question for the entire paper, this is nonspecific. Which lifestyle factors? How were they selected?

Response 5: Thank you very much for your comments. We also understand that the explanation is insufficient, as you pointed out. Therefore, we have revised the text as follows.

The purpose of this paper is to estimate QALYs using microsimulation from the influence of lifestyle behaviors that can be relatively improved, based on our previous research.

Methods:

Point 6: Line 57: was a sample size calculation performed? If so, what were the baseline assumptions?

Response 6: Thank you very much for your suggestion. In this study, we recruited participants from a larger sample than in our previous study and within our budget. Therefore, I am sorry to say that we did not calculate the sample size.

Point 7: Line 55: you say the survey was web-based; was it administered through email? If so, how were emails obtained? It is important to understand recruitment methods to understand whether there is selection bias.

Response 7: Thank you very much. We have modified the wording of the survey method as follows.

We conducted a survey of the general Japanese population by asking them to complete a web-based questionnaire format.

Point 8: Lines 66-81: were you asking these questions only about what they currently do, or also what they did in the past?

Response 8: Thank you for your question. In this survey, we asked about the participants' current lifestyle behaviors. However, only questions about smoking were set as options for past smoking history.

Results:

Point 9: Line 141: it is not until here that I understood you estimated QALYs only for sleep and physical activity. This should be made more clear in the abstract, and also the overall aims of the paper (you did not aim to estimate QALYs for all lifestyle factors, but only for ones found to be significant).

Response 9: Thank you very much for your suggestion. As you pointed out, our explanation was insufficient. We have revised the text in the abstract and methods section as follows.

[Abstract]

Therefore, we used microsimulation to estimate QALYs from physical activity and sleep, which were determined to be significant in the regression analysis.

[Methods]

Then, using only the significant factors from the regression model results, we calculated the expected HRQL score for each participant.

Discussion:

Point 10: Lines 158-161: it is good that your results were consistent with previous studies, but then what value is added by this new study?

Response 10: Thank you very much for your important remarks. We have added the following discussion of the value of this paper.

Therefore, we believe that the calculation of QALYs using physical activity and sleep as variables, which was also significant in this study, has a certain robustness and can contribute to future health economic evaluation and policy making.

Round 2

Reviewer 1 Report

The authors have made some worthwhile and useful changes to the presentation. I would recommend three additional areas for updating: 1) there still needs to be more explication of the microsimulation.....I still feel like I am guessing about how mortality expectations were derived and attributed to each case in the simulation. 2) there seem to me to be too many tables----I think tables 1,2  and 3 could be combined. 3) I appreciated clarifying that this was a web-based sample and in your notes from the authors you also commented on potential unknown bias in self-selected sample, but I did not see any new comment on potential bias in the text. 

Author Response

Response to Reviewer 1 Comments

Dear Reviewer 1,

We would like to thank you again for taking time out of your busy schedule to review our paper. We have carefully examined your comments and have made corrections to our paper, which we believe will enhance its value. I would be glad if you could check it again.

Point 1: I would recommend three additional areas for updating: there still needs to be more explication of the microsimulation.....I still feel like I am guessing about how mortality expectations were derived and attributed to each case in the simulation.

Response 1: Thank you very much for pointing this out. I am sorry that I did not explain it well enough. The microsimulation conducted in this study follows the following procedure.

  1. Generate 4000 lifestyle factors with age, gender, and significant differences by random numbers.
  2. QOL values are estimated using the above variables.
  3. The mortality rate follows only age and gender.
  4. Estimate lifetime QALYs from 2 and 3.

The following additions have been made to the treatment of age and gender factors and to the estimation of mortality.

In this simulation, we constructed a two-state model, which including alive and dead states to estimate lifetime QALYs, and created a hypothetical cohort including 4000 subjects based on the demographic statistics of our sample.  Then, using age, gender and only the significant factors from the regression model results, we calculated the expected HRQL score for each participant. We assumed that lifestyle factors were generated independently, and those factors were never changed until die for each subject. The long-term survival was modelled using the Japanese life table in 2018. Mortality was depending on age and gender of each subject. In the microsimulation, QALYs were discounted at a rate of 2% per year.

Point 2: there seem to me to be too many tables----I think tables 1,2 and 3 could be combined.

Response 2: Thank you very much for your valuable advice. We believe that there were indeed unnecessary tables included, so we have combined Table 2 and Table 3 into a new Table 2. However, Table 1 shows the demographic characteristics, and I think it is appropriate to leave it as it is, so I have left it as it is.

Point 3: I appreciated clarifying that this was a web-based sample and in your notes from the authors you also commented on potential unknown bias in self-selected sample, but I did not see any new comment on potential bias in the text.

Response 3: Thank you very much for pointing this out. In the last revision, I mentioned the possibility of selection bias at the end of the discussion, but I did not explain it well enough, so I added the following sentence.

>Since selection bias has a significant impact on the results of a survey, the results of this study may also not be based on a representative sample of the population.

Reviewer 2 Report

While the authors did revise the text where I indicated, there is very little to address the statistical and methodological concerns I had.

There are still insufficient details about the recruitment of the participants to determine whether this was a representative sample. I highly recommend you add more details as to how you recruited your participants.

I still find it very surprising that you estimate the QALYs to three decimal places in the abstract/paper and do not report the confidence intervals. Especially if they are wide, such precision is not warranted. The width of the confidence interval is an important point for understanding how accurate these QALYs are.

Author Response

Response to Reviewer 1 Comments

Dear Reviewer 2,

I would like to thank you again for your review.

Point 0: While the authors did revise the text where I indicated, there is very little to address the statistical and methodological concerns I had.

Response 0: We would like to thank you for taking time out of your busy schedule to review our paper again. We would also like to express our sincere apologies for not sufficiently correcting the points raised in the previous review. We have taken your comments seriously and made corrections, so please review our paper again.

Point 1: There are still insufficient details about the recruitment of the participants to determine whether this was a representative sample. I highly recommend you add more details as to how you recruited your participants.

Response 1: Thank you very much for your suggestion. We have added the following sentences about the characteristics of the survey firm and the survey methodology. In addition, we discussed the possibility and impact of selection bias at the end of the discussion.

>This research company has approximately 2.2 million panelists throughout Japan, and 4,000 were recruited from that target population on a first-come, first-served basis.

>Therefore, we cannot deny the possibility of a selection bias that motivated respondents were attracted. Since selection bias has a significant impact on the results of a survey, the results of this study may also not be based on a representative sample of the population.

Point 2: I still find it very surprising that you estimate the QALYs to three decimal places in the abstract/paper and do not report the confidence intervals. Especially if they are wide, such precision is not warranted. The width of the confidence interval is an important point for understanding how accurate these QALYs are.

Response 2: Thank you for your suggestions. We accept your suggestion as it enhances the value of our paper. Therefore, we have revised the QALYs listed in the text, abstract, tables, and supplementary material to one decimal place, and added 95% CIs where necessary.
